# The Value of Early Tumor Size Response to Chemotherapy in Pediatric Rhabdomyosarcoma

**DOI:** 10.3390/cancers13030510

**Published:** 2021-01-29

**Authors:** Roelof van Ewijk, Bas Vaarwerk, Willemijn B. Breunis, Reineke A. Schoot, Simone A. J. ter Horst, Rick R. van Rijn, Johanna H. van der Lee, Johannes H. M. Merks

**Affiliations:** 1Princess Máxima Center for Pediatric Oncology, 3584 CS Utrecht, The Netherlands; r.vanewijk-2@prinsesmaximacentrum.nl (R.v.E.); b.vaarwerk@amsterdamumc.nl (B.V.); R.A.Schoot-3@prinsesmaximacentrum.nl (R.A.S.); S.A.J.terHorst-3@umcutrecht.nl (S.A.J.t.H.); 2Department of Pediatric Oncology, Emma Children’s Hospital, Amsterdam UMC, University of Amsterdam, 1105 AZ Amsterdam, The Netherlands; Willemijn.Breunis@kispi.uzh.ch; 3Department of Oncology and Children’s Research Center, University Children’s Hospital, 8032 Zurich, Switzerland; 4Department of Radiology, University Medical Center Utrecht/Wilhelmina Children’s Hospital, 3584 EA Utrecht, The Netherlands; 5Department of Radiology and Nuclear Medicine, Emma Children’s Hospital, Amsterdam UMC, University of Amsterdam, 1105 AZ Amsterdam, The Netherlands; r.r.vanrijn@amsterdamumc.nl; 6Pediatric Clinical Research Office, Emma Children’s Hospital, Amsterdam UMC, University of Amsterdam, 1105 AZ Amsterdam, The Netherlands; h.vanderlee@kennisinstituut.nl; 7Knowledge Institute of the Dutch Association of Medical Specialists, 3502 GH Utrecht, The Netherlands

**Keywords:** rhabdomyosarcoma, response, prognosis, biomarker, sarcoma

## Abstract

**Simple Summary:**

Rhabdomyosarcoma is the most common soft tissue sarcoma in childhood. At diagnosis, tumor and patient characteristics determine the prognosis and subsequent treatment stratification. There are currently no early biomarkers that identify good or poor responders to chemotherapy regimens, survival being the only valid endpoint. Early tumor size response, which is assessed by imaging, could be such a marker. We performed a systematic assessment of literature to November 2020. Six studies were included describing 2010 patients; quality assessment showed methodological limitations. We conclude that there is evidence that early progressive disease is associated with poorer survival compared to patients with non-progressive disease, being either stable disease, partial, or complete response. However, for the vast majority of patients with non-progressive disease, we found no evidence that the degree of response is prognostic for survival. Therefore, the value of early tumor size response as a prognostic marker, and its translation into treatment modifications on an individual patient or trial level should be reconsidered.

**Abstract:**

Rhabdomyosarcoma is the most common soft tissue sarcoma in childhood. Results of clinical trials, with three-year event-free and overall survival as primary outcomes, often take 7 to 10 years. Identification of an early surrogate biomarker, predictive for survival, is therefore crucial. We conducted a systematic review to define the prognostic value of early tumor size response in children with IRSG group III rhabdomyosarcoma. The search included MEDLINE/EMBASE from inception to 18 November 2020. In total, six studies were included, describing 2010 patients, and assessed by the Quality in Prognosis Studies (QUIPS) instrument. Four studies found no prognostic value for tumor size response, whereas two studies reported a prognostic effect. In these two studies, the survival rate of patients with progressive disease was not separately analyzed from patients with stable disease, potentially explaining the difference in study outcome. In conclusion, our findings support that early progression of disease is associated with poorer survival, justifying adaptation of therapy. However, in patients with non-progressive disease, there is no evidence that the degree of response is a prognostic marker for survival. Because the vast majority of patients do not have progressive disease, early tumor size response should be reconsidered for assessment of treatment efficacy. Therefore, at present, early surrogate biomarkers for survival are still lacking.

## 1. Introduction

Rhabdomyosarcoma (RMS) is the most common soft tissue sarcoma in childhood and accounts for about 3–5% of all pediatric malignancies [1,2]. RMS can present at any site, most commonly in the head and neck region, the genitourinary tract, and limbs. The treatment for pediatric patients with RMS is based on a multimodality approach; at diagnosis, the majority of patients undergo a biopsy, after which induction (multidrug) chemotherapy is given, supplemented with surgery and/or radiotherapy followed by adjuvant chemotherapy. With this multimodality approach, five-year overall survival (OS) for patients with localized disease has improved to around 75% nowadays, which, with significantly lower survival in patients with metastatic disease, remains unsatisfactory [3,4].

Development and evaluation of new treatment strategies are needed to improve survival in pediatric patients with RMS. However, results of randomized clinical trials with three-year event-free survival (EFS) and OS as primary outcomes evaluating new regimens, often take 7 to 10 years to conclude. Identification of early surrogate markers of response, predictive for survival, is therefore crucial. First of all, early surrogate biomarkers facilitate a faster selection of promising new agents in phase I/II trials, therewith accelerating the transition of promising new agents or combinations into phase III trials. At the same time, agents with less promising results can be excluded early, enabling an earlier introduction and evaluation of other more promising agents. Secondly, early surrogate biomarkers could also identify patients at high risk for relapse. If we are able to identify patients at high risk for relapse at an early phase, treatment could be intensified or innovative systemic and local treatment strategies could be introduced to improve outcomes [5,6].

In European treatment protocols of the “European *paediatric* Soft tissue sarcoma Study Group” (E*p*SSG) and the “German Cooperative Soft Tissue Sarcoma” (CWS), early tumor size response was measured after 2–3 cycles of chemotherapy by conventional imaging (CT or MRI). Subsequent treatment was adjusted based on response. This implies that in patients with an insufficient response (tumor volume reduction < 33%), the chemotherapy regimen was changed to second-line chemotherapy. In contrast, patients treated according to “North-American Children’s Oncology Group” (COG) protocols were only switched to second-line chemotherapy in case of progression of disease under therapy [4,7]. This contrast merely reflects a historical difference, instead of being based on available evidence.

Therefore, the goal of this systematic literature review was to assess whether early tumor size response by radiological assessment is a surrogate marker of survival in the treatment of children with localized rhabdomyosarcoma and whether it should be used to guide subsequent therapy regimens.

## 2. Materials and Methods

The protocol was registered on PROSPERO (2017: CRD42017036060) and the Preferred Reporting Items for Systematic Reviews and Meta-Analyses (PRISMA) guidelines were used as guidance for reporting [8,9].

### 2.1. Search Strategy and Study Selection

We searched the databases of MEDLINE and EMBASE from inception to 18 November 2020, without restrictions. The electronic search strategy was developed and executed by a medical librarian. Search terms are described in Appendix A. Reference lists of included articles were checked for additional studies. The following inclusion criteria were defined: (1) pediatric patients with “Intergroup Rhabdomyosarcoma Studies clinical Group” (IRSG) III, which is defined as macroscopic residual disease at the start of chemotherapy, either after incomplete primary surgery or initial biopsy [9], histologically proven RMS, (2) tumor size response assessment after 2–4 cycles of induction chemotherapy by conventional imaging (MRI or CT), and (3) the prognostic value of early tumor size response for survival was assessed with outcomes being event-/failure-free (EFS/FFS) and overall survival (OS) after at least three years. Cohort studies, either in isolation or as part of randomized controlled trials and controlled clinical trials, were eligible for inclusion. All articles identified in the literature search were screened for titles and abstracts, followed by the full-text screening of selected articles, by two reviewers (Vaarwerk, B. and van Ewijk, R.) independently. Discrepancies between reviewers were resolved by discussion until consensus was reached or consultation of a third reviewer (Merks, J.H.M.). Studies were screened and evaluated using Covidence systematic review software (Veritas Health Innovation, Melbourne, Australia).

### 2.2. Data Extraction and Quality Assessment

Two reviewers independently extracted data using a predefined form, see Appendix B. Risk of bias of the included studies was assessed independently by two reviewers (Vaarwerk, B and van Ewijk, R.; the study of Vaarwerk et al. was appraised by van Ewijk, R. and Schoot, R.A.) using the Quality in Prognosis Studies (QUIPS) instrument, designed to assess the risk of bias for prognostic studies [10]. The QUIPS instrument consists of six domains—study participation, study attrition, prognostic factor measurement, outcome measurement, study confounding, and statistical analysis and reporting. Discrepancies between reviewers were resolved by discussion until consensus was reached or consultation of a third reviewer (Merks, J.H.M.).

### 2.3. Data Synthesis

Based on clinical heterogeneity (protocol, adjustment of therapy based on response evaluation, and patient population (e.g., age of diagnosis, percentage alveolar histology)) we decided not to perform a meta-analysis.

## 3. Results

The electronic search was performed on 18 November 2020 resulting in a total of 3189 records, see Figure 1. After the removal of duplicates, 2287 records were screened on title and abstract. We excluded 2217 records after the screening of titles and abstracts for the following reasons: review articles, editorials or letters, or case reports, studies not performed in participants with IRSG III RMS, or no assessment of response. We evaluated 70 reports in full-text. Of those, 64 reports were excluded for the following reasons: studies not performed in participants with IRSG III RMS, a replicative publication of data (only the most recent report describing the full cohort of IRSG III RMS patients was included), no assessment of response as a prognostic factor, or no early response evaluation as defined in our methods. After full-text evaluation, six studies were included in this review.

### 3.1. Study Characteristics

Six studies were included (Table 1), describing a total of 2010 patients of which 40% were female and with a predominance of the embryonal RMS subtype (77%) [10,11,12,13,14,15]. Two studies only included patients with embryonal rhabdomyosarcoma.

Five studies were retrospective analyses of prospectively collected data of multicenter cohorts, and one was a retrospective single-center cohort study. None of the studies were primarily designed to address early tumor size response evaluation, but data was collected to allow for retrospective assessment as part of the studies. The period of enrolment ranged from 6 to 25 years. Study sample sizes ranged from 62 to 529 patients with IRSG Group III RMS. Characteristics of patients included in the separate studies are reported in Table 2. Induction chemotherapy differed per study protocol; in general, it comprised a combination of alkylators (cyclophosphamide or ifosfamide), vincristine, and dactinomycin, often complemented with other agents.

### 3.2. Risk of Bias

Table 3 presents the results of the QUIPS risk of bias assessment. In summary, all studies were found to have methodological limitations. Study participation and attrition were generally good. Ermoian et al. [15] only included patients with orbital embryonal rhabdomyosarcoma, therefore the risk of participation bias was considered moderate. In the papers by Burke et al. [11], Dantonello et al. [10], Rosenberg et al. [13], and Vaarwerk et al. [14] baseline characteristics of the excluded patients were not described; therefore the risk of bias was considered moderate. Ferrari et al. only reported response assessment in 108 of 205 included patients [12]. Because patient and treatment characteristics of these 108 patients with available response assessment were not specified, the risk for study attrition bias was considered high for this study.

Response assessments were based on reports from local radiologists; none of the studies performed a central review of the tumor size response, nor was the experience of the radiologist described in the studies, potentially contributing to bias on the prognostic factor measurement domain. However, five out of the six included studies were part of multicenter international studies with concomitant guidelines on response assessment, therefore we judged the risk of bias low to moderate. In line with this assumption, we considered the outcome measurement bias low, as we expect this to be embedded in the multicenter study protocols, although (lost to) follow-up data were mostly not specified. Common limitations were concentrated on the study confounding domain as can be expected in observational studies; in three studies (Dantonello et al. [10], Ferrari et al. [12], Vaarwerk et al. [14]), subsequent therapy was based on the response assessment; two studies (Dantonello et al. [10], Ferrari et al. [12]) included patients with progressive disease in their analysis.

### 3.3. Findings

The results of the included studies are summarized in Table 4. Different response criteria were used to assess tumor size response by conventional imaging; three studies (Burke et al. [11], Rosenberg et al. [13] and Vaarwerk et al. [14]) used two-dimensional measurements according to WHO criteria [17], two studies used three-dimensional measurements (Dantonello et al. [10], Ermoian et al. [15]). Ferrari et al. [12] used both one-dimensional measurements (according to Response Evaluation Criteria In Solid Tumors [RECIST] criteria [18]) and three-dimensional measurements. All studies performing three-dimensional measurements calculated the volume based on the three measurements. None of these studies performed full tumor volume delineation (Dantonello et al. [10], Ermoian et al. [15], Ferrari et al. [12]).

Four studies (Burke et al. [11], Dantonello et al. [10], Ferrari et al. [12], and Vaarwerk et al. [14]) assessed response after three cycles of chemotherapy, and two studies (Ermoian et al. [15] and Rosenberg et al. [13]) evaluated response after four cycles of chemotherapy.

Response measurements were categorical in five studies (Burke et al. [11], Dantonello et al. [10], Ermoian et al. [15], Rosenberg et al. [13], and Vaarwerk et al. [14]); in the study of Ferrari et al. [12], continuous measurements were used. The definitions of outcome categories differed between studies, as illustrated in Figure 2.

In general, the vast majority of included patients showed an early tumor size response to first-line chemotherapy, with a complete response at early evaluation ranging from 11–31% in the included studies, see Table 4.

### 3.4. Prognostic Value of Early Tumor Size Response Assessment by Conventional Imaging

Data describing the prognostic value of early tumor size response are summarized in Table 4.

Burke et al. compared failure-free survival between patients with complete (CR), partial (PR), and no response (NR) in 444 patients included in the Intergroup Rhabdomyosarcoma Study IV. Patients with progressive disease at early response assessments were excluded. Five-year FFS was 75% for patients with CR, 71% for patients with PR, and 78% for patients with NR, respectively. Survival was compared between the different response groups by log-rank test, showing a *p*-value of 0.57 [11].

Dantonello et al. evaluated the prognostic value of early tumor size response in 529 patients with embryonal RMS (irrespective of the site) treated in five consecutive CWS trials. Five-year EFS for patients with partial response (PR) was 68.1% (95% confidence interval (CI): 64–72%) compared to 59.2% (95% CI: 46–72%) for patients with no response (NR), *p* = 0.03. The NR group included both patients with an insufficient response (less than 33%), stable disease, and progressive disease. Five-year OS was 76.4% (95% CI: 72–80%) for patients with PR compared to 62.6% (95% CI: 49–75%) for patients with NR, *p* = 0.004. Of note, patients with overt disease progression showed a five-year OS of 17% (95% CI: 0–47%) compared to 47% (95% CI: 24–70%) for patients with unchanged tumor (*p* = 0.03). The authors also evaluated prognostic factors for overall survival in a multivariate analysis, including early tumor size response by conventional imaging, treatment period, tumor site, age, tumor size, and T-status. The risk ratio for five-year mortality for patients with stable/progressive disease (SPD) was 4.8 (95% CI: 2.8–8.2) compared to patients with PR/objective response. The risk ratio for death for patients with NR was 2 (95% CI: 1.3–2.2) compared to patients with PR [10].

Ermoian et al. analyzed 53 patients with orbital embryonal RMS included in COG ARST0331. Five-year FFS was 100% for patients with CR and 84% (95% CI: 71–96%) for patients with PR or stable disease; the *p*-value of the log-rank test was 0.11. Five-year OS was 100% for patients with CR and 97% (95% CI: 91–100%) for PR/SD, *p* = 0.52 [15].

Ferrari et al. analyzed the predictive value of early tumor size response in 108 patients with RMS (irrespective of histology or site), including patients with progressive disease at response evaluation. The predictive value of tumor size response assessed by conventional imaging was evaluated in a multivariable Cox model, adjusting for sex and age, type of surgery, radiotherapy, histologic subtype, and nodal status. They found, irrespective of the method of measurement (either diameter or volume), tumor size response to be a significant predictor of survival [12]. The predictive accuracy of two multivariable models (with one model containing tumor size response as a decrease in maximum diameter and the other model containing tumor size response as three-dimensional reduction) was compared and no significant differences in predictive accuracy were found between the two models [19].

Rosenberg et al. analyzed patients with RMS (irrespective of histologic subtype or site) included in Children’s Oncology Group (COG) study D9803. In this analysis, 338 patients with RMS were included, and patients with progressive disease were excluded. Five-year FFS based on response was 74% (95% CI: 64–82%) for patients with CR, 75% (95% CI: 63–83%) for patients with PR, and 64% (95% CI: 47–82%) for patients with NR; the *p*-value of the log-rank test was 0.49 [13].

Vaarwerk et al. evaluated the prognostic value of tumor size response in 432 patients with RMS (irrespective of histology or site) included in the International Society of Pediatric Oncology (SIOP) Malignant Mesenchymal Tumor 95 (MMT-95) study. Patients with progressive disease (n = 7) were excluded from the analysis. Five-year FFS was 60% (95% CI: 55–65%) for patients with CR/PR, 60% (95% CI: 44–75%) for patients with OR and 69% (95% CI: 51–87%) for patients with NR, and the *p*-value of the log-rank test was 0.6. Five-year OS was 74% (95% CI: 69–79%) for patients with CR/PR, 73% (95% CI: 58–87%) for patients with OR, and 72% (95% CI: 55–90%) for patients with NR, *p* = 0.9. The prognostic value of tumor size response by conventional imaging was further evaluated in a Cox regression analysis adjusting for histology, tumor size, tumor site, nodal involvement, age, radiotherapy, and late surgery. The adjusted hazard ratios for OR and NR compared to sufficient response were 1.09 (95% CI: 0.63–1.88) and 0.81 (95% CI: 0.39–1.67), respectively for FFS, and 0.91 (95% CI: 0.47–1.76) and 1.27 (95% CI: 0.61–2.64), respectively for OS [14].

## 4. Discussion

### 4.1. Discussion of Findings

The results of this systematic review, including data from large international study groups, clearly illustrate the discordant results in the existing literature on the prognostic value of early tumor size response in pediatric RMS. There is no evidence that the degree of response at early response evaluation, except for patients with progressive disease, is a prognostic marker for survival. As the majority of patients do not have progressive disease, the use of early tumor size response should be reconsidered for individual treatment adjustments or to evaluate new treatment strategies. However, the results do illustrate that the early progression of disease impacts survival.

Studies included in this review included heterogeneous cohorts of patients with different RMS histology, tumor sites, age, and treatment strategies, reflecting clinical practice. Studies applied different inclusion criteria, heterogeneous categorization of response groups, and different statistical approaches leading to important limitations in interpretation and comparability of these studies.

Four studies included only patients with non-progressive disease at early response evaluation, where two studies included all patients. In the studies including all patients, progressive disease was not investigated as a separate group and could as such have influenced the conclusion of the studies. Dantonello et al. compared patients with good response (any patient with ≥33% tumor size response) to patients with no response (NR) and showed a significant difference in survival. The NR group included patients with an insufficient response (<33%), stable disease, and progressive disease, with a large difference in survival between these groups as mentioned in the sub-analysis [11]. Ferrari et al. investigated tumor size response as a continuous measure, including all tumor size responses in analyses [12]. Whether the significance of the measure reflects the poor prognosis of patients with progressive disease or also relates to the degree of response in responding patients was not investigated. Therefore, based on these two studies, the value of early tumor size response in non-progressive disease for the vast majority of patients remains unclear. These studies do underline the poor survival of patients with progressive disease at early tumor size response assessment, also shown in other studies, which justifies individual treatment adaptation in this category [20]. Of notice, although the overt progressive disease is mostly obvious, well-performed studies identifying optimal cut-offs for the definition of progressive disease have not been performed in pediatric RMS.

The four other studies, which included only non-progressive patients, showed no prognostic value for early tumor size response without significant survival differences for patients with a complete response and stable disease. However, interpretation of these studies is further limited by the fact that the European studies included in this review (Dantonello et al. [10], Ferrari et al. [12], and Vaarwerk et al. [14]) adapted treatment based on response. Furthermore, in the studies of Dantonello et al. [11] and Vaarwerk et al. [14] specific patients in favorable subgroups with complete response to induction chemotherapy did not receive radiotherapy as first-line therapy. 

Next to cohort selection and treatment adjustments, it is important to discuss the quality of the imaging and the method of assessment. In most studies, patients were included before 2000. The quality of the imaging has improved significantly over the last decades. Response measurements in all included studies were performed by local radiologists with different dimensional assessments, yet previous studies showed that tumor size response measurement is subject to important interobserver and intermethod variability [21,22]. Because none of the studies collected the original CT/MR imaging or performed structural central review, questions concerning the importance of interobserver and intermethod variability, experience of radiologist, and potential bias in local assessment remain unanswered. 

Finally, despite the large international cohort studies included in this review, no final conclusion can be drawn considering the prognostic value of early tumor size response in children with non-progressive RMS due to the heterogeneity and limitations of all studies. Future high-quality studies should answer this question for the different subgroups in rhabdomyosarcoma. 

### 4.2. Strengths and Weaknesses

This review is the first thorough evaluation of the available evidence on the prognostic value of early tumor size response by conventional imaging in pediatric RMS including data on 2010 patients. Extensive search and systematic evaluation without restriction on language and publication status of the currently available evidence was performed in line with our registered PROSPERO protocol [8]. Studies were quality assessed to provide an unbiased assessment of evidence. For the study of Vaarwerk et al. [14], we aimed to limit bias by performing the quality assessment by two researchers which were not involved in the evaluated study (van Ewijk, R. and Schoot, R.A.).

Unfortunately, the data are only reported descriptively in this overview because performing a meta-analysis was considered inappropriate due to large heterogeneity between included studies. This heterogeneity was caused by different methods of response measurement, different statistical methods, and differences in treatment consequences based on response assessment outcome. A limitation is that we did not try to collect all original data. 

### 4.3. Future Perspective

The search for early surrogate markers of survival, either biological or imaging biomarkers, should be part of future clinical rhabdomyosarcoma trials. In future trials, there is a need for consistent imaging guidelines developed by an imaging study group, wherein all sites use the same imaging modality and acquisition protocol to provide homogeneous acquisition, to evaluate more than the tumor size response. Guidelines should define the optimal criteria for response assessment. To evaluate these criteria there is a strong need for centralized evaluation and collection of imaging to limit inter-observer variability and to allow for (semi)-automated assessment. Centralized collection also provides the possibility for quality control and feedback to collaborating investigational sites, thereby improving imaging in pediatric rhabdomyosarcoma.

In rhabdomyosarcoma, we expect future research on prognostic imaging biomarkers will combine multiple parameters from different scan modalities. Systematic evaluation of morphological features of conventional MRI in combination with specific MR sequences, such as diffusion-weighted MR, may provide a further understanding of the chemotherapy-induced changes, in combination with metabolic activity, assessed by [18F]-fluorodeoxyglucose positron emission tomography (FDG PET) or newly developed PET tracers [23]. In line with global development in radiology we expect that advanced algorithms will support the search for the optimal response marker set. Centralized collection of multiple modalities will provide a platform for researchers specialized in advanced machine learning techniques.

## 5. Conclusions

The results of this study illustrate the discordance on the prognostic value of early tumor size response assessed by conventional imaging in RMS. Based on the results of the included studies, we conclude that the current literature shows no evidence for early tumor size response as an early surrogate marker of survival in children with non-progressive RMS. Therefore, tumor size response should not be used to adapt subsequent treatment, except for patients showing progression of disease at early assessment.

## Figures and Tables

**Figure 1 cancers-13-00510-f001:**
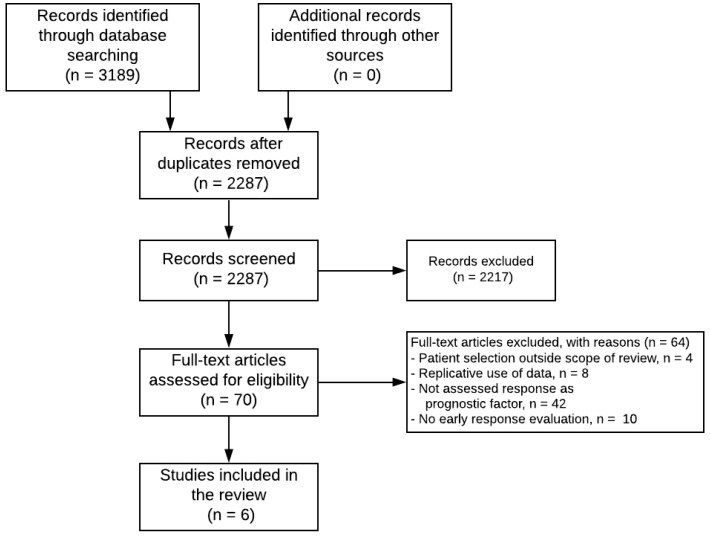
Flowchart of study inclusion.

**Figure 2 cancers-13-00510-f002:**
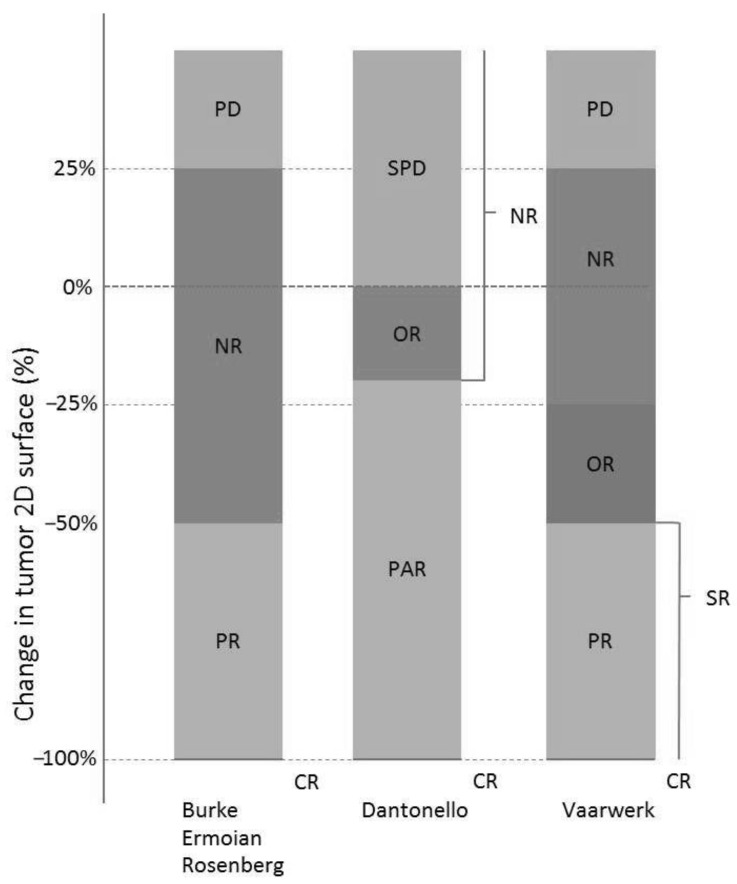
Grouping of response categories for the different studies. Categories expressed as change in tumor size, as measured in two-dimensional measurements %. Studies describing a different response measurement (i.e., one-dimensional or three-dimensional measurement) were recalculated. Abbreviations: CR, complete response; NR, non-response; OR, objective response; PD, progressive disease; PR/PAR, partial response; SPD, stable/progressive disease; SR, sufficient response. In three studies (Dantonello et al. [10], Ferrari et al. [12], and Vaarwerk et al. [14]) treatment after response assessment was adjusted in case of insufficient response based on the tumor size response.

**Table 1 cancers-13-00510-t001:** Summary of the studies included in this systematic review.

Study (year)	Country	Study Design	Enrolment Period	No. of Patients Included in Analysis	Reason for Excluding Patients from Response Assessment Analysis
Burke et al. (2007)	Multinational	Multicenter retrospective cohort study	1991–1997	444	Off therapy before completion of induction therapy/no response assessment (n = 49) Other histology than ERMS or ARMS (n = 41) Start date of RT could not be determined (n = 14)
Dantonello et al. (2015)	Multinational	Multicenter retrospective cohort study; 5 consecutive trials	1980–2005	529	In total n = 229 excluded: No documented measurement at the correct evaluation point Relevant tumor part removed at primary surgery Surgery/radiotherapy prior to evaluation of response
Ermoian et al. (2017)	USA	Multicenter retrospective cohort study	2004–2010	53	PD before week 12 evaluation (n = 2) Insufficient or missing week 12 evaluation (n = 7)
Ferrari et al. (2010)	Italy	Single-center retrospective cohort study	1982–2008	205 (108 with response assessment)	In total n = 216 excluded: Metastatic disease Missing information on initial tumor size Radiologic diameter and volume not assessed
Rosenberg et al. (2014)	Multinational	Multicenter retrospective cohort study	1999–2005	338	Other histology than ERMS or ARMS (n = 90) Not IRS group III (n = 139) No response measurement documented (n = 20) PD at response assessment (n = 6)
Vaarwerk et al. (2017)	Multinational	Multicenter retrospective cohort study	1995–2003	432	In total n = 194 excluded: Unknown tumor size (n = 64) No response evaluation or at wrong time (n = 116) Tumor response was not evaluable (n = 5) Progressive disease at response assessment (n = 7) Lost to follow-up (n = 2)

Abbreviations: ARMS, alveolar rhabdomyosarcoma; ERMS, embryonal rhabdomyosarcoma; IRSG, Intergroup Rhabdomyosarcoma Group post-surgical staging; PD, progressive disease; RT, radiotherapy.

**Table 2 cancers-13-00510-t002:** Patient characteristics for included studies.

Characteristics	Burke et al. 2007	Dantonello et al. 2015	Ermoian et al. 2018	Ferrari et al. 2009	Rosenberg et al. 2014	Vaarwerk et al. 2017	Total
	N (%)	N (%)	N (%)	N (%)	N (%)	N (%)	N (%)
**Total number of patients**	444	529	62	205	338	432	2010
**Sex**							
Female	189 (43)	NR	24 (39)	72 (35)	127 (38)	184 (43)	596 (40)
Male	255 (57)	NR	38 (61)	133 (65)	211 (62)	248 (57)	885 (60)
**Age, years**							
≤10	327 (74)	450 (85)	***	103 (50)	249 * (74)	345* (80)	1474 (76)
>10, ≤14	71 (16)	79 (15)		40 (20)	89 * (26)	87* (20)	366 (19)
>14	49 (11)			62 (30)			111 (6)
**Tumor site**							
Extremity	40 (9)	16 (3)		24 (12)	49 (15)	47 (11)	176 (9)
GU-nonbladder/prostate	32 (7)	28 (5)		51 (25)	43 ** (13)	26 (6)	180 (9)
GU-bladder/prostate	58 (13)	91 (17)		13 (6)		66 (15)	228 (11)
PM	178 (40)	194 (37)		50 (24)	155 (46)	134 (31)	711 (35)
HN-nPM	20 (4)	31 (6)		34 (17)	7 (2)	43 (10)	135 (7)
Orbit	47 (11)	72 (14)	62 (100)	NS	12 (4)	59 (14)	252 (13)
Pelvis/trunk	NR	NR		33 (16)	42 (12)	NS	75 (4)
Other	69 (15)	97 (18)			30 (9)	57 (13)	253 (13)
**Histological subtype**							
Alveolar	103 (23)			61 (30)	132 (39)	144 (33)	440 (22)
Embryonal	323 (71)	529 (100)	62 (100)	136 (66)	206 (61)	288 (67)	1544 (77)
NOS	18 (4)			8 (4)			26 (1)
**Tumor size, cm**							
≤5	187 (42)	212 (40)	60 (97)	78 (38)	139 (41)	217 (50)	893 (44)
>5	255 (58)	263 (50)	1 (2)	127 (62)	199 (59)	215 (50)	1060 (53)
Unknown	54 (10)						57 (3)
**T status**							
T1	140 (32)	146 (28)		66 (32)	152 (45)	152 (35)	656 (34)
T2	302 (69)	370 (70)		139 (68)	185 (55)	272 (63)	1268 (65)
Unknown	2	13 (2)				8 (2)	23 (1)
**N status**							
N0	332 (79)	437 (83)		158 (77)	274 (81)	347 (80)	1548 (80)
N1	86 (21)	62 (12)		47 (23)	64 (19)	71 (16)	330 (17)
Unknown	26	30 (6)				14 (3)	70 (4)

Abbreviations: GU, genitourinary; HN-nPM, head and neck nonparameningeal; N0, no evidence of lymph node involvement; N1, evidence for lymph node involvement; NR, not reported; NOS, not otherwise specified; PM, parameningeal; T1, tumor confined to the organ or tissue of origin; T2, tumor not confined to the organ or tissue of origin. * Definition in Rosenberg et al. 2014 and Vaarwerk et al. 2017, Age < 10 and Age ≥ 10. ** Genitourinary not further specified. *** Cut off at six years.

**Table 3 cancers-13-00510-t003:** Quality assessment, based on Quality in Prognosis Studies (QUIPS) [16] instrument, assessing the prognostic value of early tumor size response to chemotherapy in pediatric rhabdomyosarcoma.

Study (year)	Study Participation	Study Attrition	Prognostic Factor Measurement	Outcome Measurement	Study Confounding	Statistical Analysis Reporting
Burke et al. (2007)	Low	Moderate	Low	Low	Moderate	High
Dantonello et al. (2015)	Low	Moderate	Moderate	Low	High	Moderate
Ermoian et al. (2018)	Moderate	Low	Moderate	Low	Low	Low
Ferrari et al. (2010)	Moderate	High	Low	Moderate	High	High
Rosenberg et al. (2014)	Low	Moderate	Low	Low	Moderate	Moderate
Vaarwerk et al. (2017)	Low	Moderate	Moderate	Low	Moderate	Low

**Table 4 cancers-13-00510-t004:** Survival for response groups of included studies.

Study (year)	Response Group, n (%)	Outcomes Based on Response		
		5-yr FFS/EFS ^¥^	5-yr OS ^¥^	Multivariable Analysis
Burke et al. (2007) *^§^	CR: n = 94 (21%)	CR: 75%		
	PR: n = 248 (56%)	PR: 71%		
	NR: n = 102 (23%)	NR: 78%		
		*p* = 0.57		
Dantonello et al. (2015)	PAR: n = 470 (89%)	PAR: 68.1% (64–72%)	PAR: 76.4% (72–80%)	Risk ratio of death:
	NR: n = 59 (11%)	NR: 59.2% (46–72%)	NR: 62.6% (49–75%)	PAR + OR = 1, SPD = 4.8 (2.8–8.2) ^¥^
		*p* = 0.03	*p* = 0.004	Risk ratio of death:
				PAR = 1, NR = 2 (1.3–3.2) ^¥^
Ermoian et al. (2017)	CR: n = 15 (28%)	CR: 100%	CR: 100%	
	PR: n = 31 (59%)	PR/SD: 84% (71–96%)	PR/SD: 97% (91–100%)	
	SD: n = 7 (13%)	*p* = 0.11	*p* = 0.52	
Ferrari et al. (2010) ^#^	Response evaluated as continuous variable			Response significant predictor of survival (Wald test *p* < 0.001 for diameter and volume). V measure: 0.300 for diameter, 0.323 for volume.
Rosenberg et al. (2014) *^§^	CR: n = 95 (28%)	CR: 74% (64–82%)		
	PR: n = 193 (57%)	PR: 76% (63–83%)		
	NR: n = 50 (15%)	NR: 64% (47–82%)		
		*p* = 0.49		
Vaarwerk et al. (2017) *^§^				Hazard ratios FFS: ^$^
	SR: n = 261 (85%)	SR: 60% (55–65%)	SR: 74% (69–79%)	SR: 1, OR: 1.09 (0.63–1.88) ^¥^
	OR: n = 27 (9%)	OR: 60 (44–75%)	OR: 73% (58–87%)	NR: 0.81 (0.39–1.67) ^¥^
	NR: n = 19 (6%)	NR: 69% (51–87%)	NR: 72% (55–90%)	Hazard ratios OS: ^$^
		*p* = 0.6	*p* = 0.9	SR: 1, OR: 0.91 (0.47–1.76) ^¥^
				NR: 1.27 (0.61–2.64) ^¥^

Abbreviations: CR, complete response; EFS, event-free survival; FFS, failure-free survival; HR, hazard ratio; IRS, Intergroup Rhabdomyosarcoma Group post-surgical staging; NR, non-response; OR, objective response; OS, overall survival; PAR/PR, partial response; RMS, rhabdomyosarcoma; RT, radiotherapy; SD, stable disease; SPD, stable/progressive disease; SR, sufficient response; yr, years. ^¥^ In brackets 95% confidence interval, if reported. ^¥^ statistical test not specified. ^#^ according to RECIST criteria [17]. * according to WHO criteria [16]. ^$^ Adjusted for histology, tumor size, tumor site, nodal involvement, age, radiotherapy, and post-chemotherapy surgery. ^§^ Survival for response groups with alveolar histology is summarized in Appendix C.

## Data Availability

The datasets used and/or analyzed during the current study are available from the corresponding author on reasonable request.

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
