# Peer review of "The Value of Early Tumor Size Response to Chemotherapy in Pediatric Rhabdomyosarcoma"

_cancers, 2021, doi:10.3390/cancers13030510_

Round 1
Reviewer 1 Report
Introduction
Was it the original idea to concentrate on tumor size only and not on other factors - later on you state in materials and methods that age of dg, percentage of alveolar RMS) perhaps prevented this
Materials and methods
1) Inclusion criteria: I am always a bit confused By the IRSG group III ,
What is meant by primary surgery : Does macroscopic residual disease after primary surgery mean a) macroscopic residual disease after surgery following induction chemotherapy or b) macroscopic residual disease after ´upfront surgery´ie surgery without preceding chemo or both - I think this should be made clear
2) There was a fairly consistent percentage of alveolar RMS in four studies (23 - 39%) , why not assess these separately.
Also N1 status was consistent 16 - 23% in six studies
Results
A summary table would be easier than Table 4 as such, now Table is a bit hard to read, could you not leave out response evaluation criteria and timing response which are already mentioned in the text and concentrate on response groups and outcome, the Table would be more compact and convey the message just as well.
Discussion
The discussion is compact and to the point. Perhaps you could discuss shortly whether there are any other factors in tumor management that may profit from the assessment of tumor response - it is clear that intensifying treatment (chemoptherapy ?) cannot be based on tumor response in non-progressing tumors - there is no benefit in survival . How about doing early redo surgery, radiotherapy, brachytherapy.
Conclusion
I agree with the conclusion
Author Response
Dear reviewer,
Thank you very much for your valuable feedback and questions improving our systematic review. Please allow us to respond.
- Original idea:
Dear reviewer, yes, the original idea was to concentrate on tumor size only, as prespecified in our PROSPERO protocol (CRD42017036060). As planned we included all studies that fulfilled the inclusion criteria, where the study population was defined as children aged 0-18 years with IRSG group III RMS, irrespective of histological subtype. In section 2.3, page 3, line 117, we refer to the patient population (age, % of alveolar subtype). Primarily, we considered not to perform a meta-analysis due to the clinical heterogeneity of the included studies. As the included studies also presented heterogeneous patient populations we aimed to underline this heterogeneity, which should also be considered when aiming to compare these studies; factors we know that correlate with survival.
- IRSG group III
IRSG group III is defined as: “A localized tumor, with macroscopic residual disease after grossly incomplete removal, or biopsy only”. The IRSG group is classifying before start of chemotherapy. We agree clarifying this improves reading for the general audience. Page 2, line 94. We changed: “which is defined as macroscopic residual disease after primary surgery or initial biopsy”, to “which is defined as macroscopic residual disease at start of chemotherapy, either after incomplete primary surgery or initial biopsy”
- Sub-grouping of alveolar / nodal status in four studies
We did not specify subgroup analyses in our preregistered PROSPERO protocol, therefore, we did not perform subanalyses for alveolar histology, nor for nodal status. Most importantly, the clinical study heterogeneity limited a meta-analysis in our opinion, independent of histology or nodal status. To provide an overview of available survival data for nodal status and alveolar histology we added a supplement that provides an overview. See Appendix C “Survival and response specified for alveolar histology”, see page 14, line 468.
- Table 4
Dear reviewer we agree with your statement that we can leave out response evaluation criteria and timing of response in this table to provide a more clear overview. Please find the revised table on page 8, line 268. We deleted three columns: response evaluation criteria, timing of response and treatment adjustment.
- Discussion
Dear reviewer, thank you your compliment for our discussion. You wonder whether we could elaborate on potential other strategies after early response evaluation. As current evidence does not clearly provides subgroups with better or worse survival, other than patients with progressive disease, we do not consider it of added value to speculate on other treatment strategies after response assessment in this specific manuscript.
Reviewer 2 Report
Table 1: Please check the correct numbers in the reference of Ferrari's study, 205 patients included and 216 excluded. Is correct the huge number of patients excluded? This contrasts with the other studies cited in this table.
In the discussion the authors state that the reduction in size is not of prognostic value in RMS. This should be questioned and this analysis do not discard not prove this concept. This is speculative and should be clarified in future studies. It would be more appropriate to say that in the group of patients without reduction in the size of the tumor after the initial cycles of chemotherapy we need to use other methods such as DW-MRI or FDG-PET to determine the tumor response.
Author Response
Dear reviewer,
Thank you very much for your valuable feedback and questions improving our systematic review. Please allow us to respond.
- Table 1
Dear reviewer yes 216 patients were excluded of the original cohort. In the first paragraph of “Patients and methods” in the paper by Ferrari et al it states: “From a series of 421 consecutive patients with RMS up to 21 years of age at the time of diagnosis, all included in the database of the Pediatric Oncology Unit at the Istituto Nazionale Tumori, Milan, Italy, we selected patients with nonmetastatic disease and full information on initial tumor size, assessed radiologically in terms of both diameter and volume. The resulting study population consisted of 205 previously untreated patients with RMS, seen from January 1982 (when CT and/or MRI became routinely available at our institute) to January 2008. To analyze tumor response, we selected a subset from this sample comprising 108 cases with complete details on tumor size (diameter and volume) both at diagnosis and after three cycles of induction chemotherapy.”
In the paper by Ferrari et al. the cohort of 205 patients is included as described in table 2 in the paper of Ferrari et al. We therefore state that 216 patients are excluded. In total 421 – 105 = 316 patients did not have full data available of response. We consider this clear from table 1.
- Discussion
Dear reviewer, thank you for addressing this point. We agree with you that your statement that reduction in size is not of prognostic value in RMS should be questioned and that future, centrally analyzed, studies should provide better evidence. Therefore, we state in the first paragraph of our discussion, page 9, line 280, the following: “There is no evidence that the degree of response at early response evaluation, except for patients with progressive disease, is a prognostic marker for survival.”. The statement that there is no evidence for is not equivalent to the statement that there is evidence that the degree of response is not prognostic. In line with your comment we therefore stated that early tumor size response should be reconsidered, we did purposely did not state that is should discarded. In absence of disease we therefore used the words ‘at the moment’ in our conclusion on early tumor size response (conclusion, line 372).
Reviewer 3 Report
Brief summary: The authors make a systematic review to define the prognostic value of early tumor size response in children with IRSG group III rhabdomyosarcoma. The paper is well written.
The goal of the systematic literature review was to assess whether early tumor size response by radiological assessment is a surrogate marker of survival in the treatment of children with a localized rhabdomyosarcoma and whether it should be used to guide subsequent therapy regimens.
The search strategy and study selection, Data extraction and quality assessment, Data synthesis, Study characteristics, Risk of bias, and Findings were well synthesized and summarized in figures and tables.
Data describing the prognostic value of early tumor size are described for each of 6 included articles.
In the Discussion paragraph, the authors clearly illustrate the discordant results in the existing literature on the prognostic value of early tumor size response in pediatric RMS and highlight despite the large international cohort studies included in this review, and highlight that no final conclusions can be drawn considering the prognostic value of early tumor size response in children with non-progressive RMS, due to the heterogeneity and limitations of all studies.
The Conclusions are coherent with Discussion and data analysis.
Main points:
The sentence, "we conclude that there is evidence to use early tumor size response as an early marker of response in case of progressive disease", is a sentence that seems to contradict the one that follows.
The sentence "However, for the vast majority of patients with non-progressive disease, we found no evidence that the degree of response is prognostic for survival." needs more explanations.
It is not easy to understand. The sentence suggests that there are other factors affecting survival, regardless of early tumor size response.
There are some contradictory/unclear sentences if we compare the Summary, the Abstract and the Conclusions. I suggest trying to harmonize the three paragraphs.
and it’s translation (line 29)
suggestion: and its translation (line 29)
Author Response
Dear reviewer,
Thank you very much for your valuable feedback and questions improving our systematic review. Please allow us to respond.
- Progressive disease – other factors
Dear reviewer thank you for providing us the opportunity to clarify this section better. In the simple summary, line 26 we changed the sentence “We conclude that there is evidence to use early tumor size response as an early marker of response in case of progressive disease.” to “We conclude that there is evidence that early progressive disease is associated with poorer survival compared to patients with non-progressive disease, being either stable disease, partial or complete response.
We think this is better in line with Abstract, line 42: “In conclusion, our findings support that early progression of disease is associated with poorer survival, justifying adaptation of therapy. However, in patients with non-progressive disease, there is no evidence that the degree of response is a prognostic marker for survival.”
In our conclusion we provide similar statements for patients with either non-progressive or progressive disease. We agree that the sentence (conclusion, line 374) “This means that, at the moment, we cannot use early tumor size response to determine the efficacy of treatment.” is not in agreement with the statement for progressive disease. Therefore, we deleted this sentence aiming to harmonize the simple summary, abstract and conclusion.
- Textual suggestion was adjusted.
Reviewer 4 Report
I applaud the authors on a comprehensive, clear and well-presented systemic review on an important topic. I agree with their premise that it is important to determine whether any intermediate predictors of response correlate with survival in RMS. I believe their methodology is rigorous and that they appropriately and comprehensively summarize existing data on this topic. Unfortunately, as the authors rightly acknowledge, significant heterogeneity in the available data limits the ability to draw any meaningful conclusions even after aggregating available data.
The authors appropriately delineate the limitations of their study, but I wonder if there are differing ways to interpret the findings. If patients with early signs of treatment response (as measured by tumor measurement) have equivalent outcomes while avoiding therapy intensification, shouldn’t that be considered a positive result? Perhaps this point should be made a little more strongly in the discussion.
In the Rosenberg study there was a trend towards significance in the subset of patients with ARMS. Do the authors have the necessary data to comment on correlation between treatment response and survival in the subset of patients with ARMS (beyond what is already alluded to in the Cox regression analyses referenced in a few of the studies)?
Minor comments:
- Is it appropriate to call an imaging finding a “biomarker”? In other places you use the term “surrogate marker”, which I think is better. Or “intermediate marker”
- In the Discussion, Future perspectives section, it may be worth referencing recent data showing no predictive value of PET response in RMS (PMID 33340280).
- Was this entire study really completed in 4 weeks after performing an initial literature search on November 18th?
Author Response
Dear reviewer,
Thank you very much for your valuable, positive feedback and questions improving our systematic review. Please allow us to respond.
- Results after treatment intensification
Dear reviewer, thank you very much for your question. We agree that the influence of treatment intensification could bias the results of the early response assessment. However, this would only be the case if treatment intensification led to improved outcome. Oberlin et al published the survival data of the randomization in high risk patients between standard therapy and treatment intensification (3 versus 6 drugs), which did not show significantly different survival (Randomized comparison of intensified six-drug versus standard three-drug chemotherapy for high-risk nonmetastatic rhabdomyosarcoma and other chemotherapy-sensitive childhood soft tissue sarcomas: long-term results from the International Society of Pediatric Oncology MMT95 study. J Clin Oncol. 2012 Jul 10;30(20):2457-65). Therefore, we consider similar outcome after treatment intensification not a positive result in this specific cohort and did not adjust our manuscript for this. In general, we agree with you that treatment adjustments should be taken into account.
- Significant subset alveolar RMS Rosenberg
Dear reviewer, unfortunately in the publication by Rosenberg et al. this is not further described.
- Biomarker
O’Connor et al. (Imaging biomarker roadmap for cancer studies. Nat Rev Clin Oncol. 2017 Mar;14(3):169-186.) stated that ‘A biomarker is a “defined characteristic that is measured as an indicator of normal biological processes, pathogenic processes or responses to an exposure or intervention, including therapeutic interventions’. Objective response, as measured on imaging, is mentioned as a biomarker. We have reviewed all our sentences where we use the term ‘marker’. We changed marker to biomarker in line 35, line 64 and line 68 where marker was not followed by the term ‘of response’ as we consider the use of both in one sentence a pleonasm. We added this reference in line 71 so readers can find this publication defining a biomarker.
- Addition of reference
Thank you for your addition, we added the reference to this recent publication.
- Date of search
No the search was performed previously. We updated the search to the 18th of November to provide up-to-date evidence.
Reviewer 5 Report
The study reports the current state of art of early tumor size response, as early predictive surrogate marker, assessed by imaging, to chemotherapy regimens in paediatric Rhabdomyosarcoma.
The paper is interesting and useful to give to readers a clear and concise description of the state of art for early tumor size evaluation of patients suffering from Rhabdomyosarcoma.
The Authors concluded that the use of early tumor size response should be reconsidered for individual treatment adjustments or to evaluate new treatment strategies but early surrogate biomarkers for survival are still lacking, except for patients with progressive disease.
Since the principal weak of the study is heterogeneity of the studies included in the review (as discussed in the limitations section) it would be interesting, if possible, whether the Authors would present the results of their systematic review of literature, enlarging the studies cohort including article with similar patient sample in term of therapy, statistical approaches and radiological assessment.
Abstract:ok;
Introduction: clear and balanced;
Materials and Methods: methods are well described
Results: The results are clearly described;
References: too short in my opinion
Author Response
Dear reviewer,
Thank you very much for your valuable, positive feedback and questions improving our systematic review. Please allow us to respond.
- Enlarging study cohorts
Dear reviewer, thank you for your feedback. We have performed an extensive search aiming to include all relevant articles reporting radiological assessment currently available. If we understand your question well that you ask whether we would like to enlarge our inclusion criteria for a wider search of studies, we believe the strength of our research is that we prespecified our inclusion criteria in the registered PROSPERO protocol. Therefore, we think we should not change our inclusion criteria at this moment. If you mean whether we would be open to update our review when new results appear, yes we would like be open for such initiative, although we believe current global initiatives of data harmonization might provide better and more in depth analysis of the data available and its heterogeneity.
- References
We aimed to include all relevant references. Due to the valuable feedback of all reviewers we added 2 references.